# S-Nitrosylation: An Emerging Paradigm of Redox Signaling

**DOI:** 10.3390/antiox8090404

**Published:** 2019-09-17

**Authors:** Veani Fernando, Xunzhen Zheng, Yashna Walia, Vandana Sharma, Joshua Letson, Saori Furuta

**Affiliations:** Department of Cancer Biology, University of Toledo Health Science Campus, 3000 Arlington Ave., Toledo, OH 43614, USA; VeaniRoshale.Fernando@rockets.utoledo.edu (V.F.); Xunzhen.Zheng@utoledo.edu (X.Z.); Yashna.Walia@rockets.utoledo.edu (Y.W.); vandana988@gmail.com (V.S.); Joshua.Letson@utoledo.edu (J.L.)

**Keywords:** NO, S-nitrosylation, NOS, ROS, antioxidant, redox regulation

## Abstract

Nitric oxide (NO) is a highly reactive molecule, generated through metabolism of L-arginine by NO synthase (NOS). Abnormal NO levels in mammalian cells are associated with multiple human diseases, including cancer. Recent studies have uncovered that the NO signaling is compartmentalized, owing to the localization of NOS and the nature of biochemical reactions of NO, including S-nitrosylation. S-nitrosylation is a selective covalent post-translational modification adding a nitrosyl group to the reactive thiol group of a cysteine to form S-nitrosothiol (SNO), which is a key mechanism in transferring NO-mediated signals. While S-nitrosylation occurs only at select cysteine thiols, such a spatial constraint is partially resolved by transnitrosylation, where the nitrosyl moiety is transferred between two interacting proteins to successively transfer the NO signal to a distant location. As NOS is present in various subcellular locales, a stress could trigger concerted S-nitrosylation and transnitrosylation of a large number of proteins involved in divergent signaling cascades. S-nitrosylation is an emerging paradigm of redox signaling by which cells confer protection against oxidative stress.

## 1. Introduction

In the past decades, nitric oxide (NO) has garnered an increasing amount of interest with regard to its impact on many human diseases. Since its discovery by Furchgott et al. as “the endothelium-derived relaxing factor (EDRF)” over 30 years ago [1], NO has become one of the most studied subjects in the field of biomedical sciences. NO was named “the Molecule of Year” in 1992, and its discovery led to the 1998 Nobel Prize of Furchgott et al. Despite the original conception of NO as a freely diffusible gas [2], NO production and signaling are rather compartmentalized and spatially regulated. This is owing to the fact that all the isoforms of NO synthases (NOS1-3) are mostly localized at the membrane and organelles [3,4,5], and because NO produced there is quickly (<0.1 sec) consumed by reacting with the proximal molecules [5,6,7]. On one hand, NO reacts with molecular oxygen to become nitrogen oxide, which then reacts with a transition metal or cysteine thiol to form a nitrosyl adduct (metal nitrosylation or S-nitrosylation, respectively). The nitrosyl group could be further transferred to another protein in a distant site (transnitrosylation) to propagate NO signaling. Such effects of NO are said to be “direct” and involved in many anti-oxidant mechanisms as well as other biochemical activities [8]. On the other hand, in the presence of high levels of reactive oxygen species (ROS), such as superoxide, NO reacts with ROS and forms a strong oxidant, reactive nitrogen species, which in turn reacts with and oxidatively damages a variety of biological molecules. This mode of NO effect is said to be “indirect” and involved in “pro-oxidant” signaling of NO [8].

Initial NO studies focused on the NO signaling pathway in specialized tissues such as neurons, muscles, endothelia and immune cells. For example, in the brain NO controls brain-blood flow and other processes that lead to normal brain function. In the circulatory system, NO controls leucocyte adhesion, pro-inflammatory signaling, platelet aggregation, and angiogenesis [9]. Such diverse functions of NO are mediated by the “classical” and “non-classical” schemes. In the classical NO signaling, NO binds the heme iron of guanylyl cyclase (sGC) to induce production of cGMP, which then activates the cGMP-dependent protein kinase (PKG) pathway [10]. Notwithstanding this classical scheme, recent studies have unveiled a wealth of “non-classical” NO signaling that mediates pleiotropic functions in diverse tissues/organs. A major mechanism of non-classical NO’s functions is S-nitrosylation, an NO-dependent covalent modification of a cysteine thiol [11]. S-nitrosylation triggers a change in the protein structure, which could alter protein-protein interactions and enable further post-translational modifications such as phosphorylation, acetylation, ubiquitination, and disulfide bond formation [11,12,13,14]. Furthermore, in the presence of a low level of ROS, S-nitrosylation could not only scavenge NO to prevent it from reacting with ROS, but also protect cysteine thiols against ROS-mediated oxidation [sulfonic acid (RSO3H) formation] [15]. At very high levels of ROS, however, NO could react with ROS to form reactive nitrogen species as discussed above [8,16].

S-nitrosylation could regulate >3000 proteins involved in a multitude of biological processes, including protein stability/turnover, steroid synthesis, transcription regulation, DNA damage repair, cellular growth/differentiation, apoptosis and redox regulation. Thanks to various subcellular locales of NOS, a stress could induce concerted S-nitrosylation and transnitrosylation of proteins involved in divergent signaling cascades. Dysregulated S-nitrosylation has been implicated in the incidence and progression of different diseases [11,15,17,18,19]. In this review, we will provide the recent advances in our understanding of how S-nitrosylation exerts redox regulation on cells and tissues. We will especially focus on cellular locales which are equipped with various mechanisms that produce or fight against ROS, such as the mitochondrion, nucleus, and extracellular environment. We will then discuss how dysregulated S-nitrosylation would lead to disease conditions.

## 2. NO and Its Roles in Redox Regulation

NO is a reactive, gaseous signaling molecule with a half-life of 0.09–2 seconds [6]. NO is produced by nitric oxide synthase (NOS) using amino acid L-arginine as the substrate in a wide range of tissues and is involved in pleiotropic functions. Most NO studies have focused on its role in specialized cells, including neurons, muscles, endothelia, and immune cells. In these cells, NO mediates neurotransmission, muscle and vessel dilation, and pro-inflammatory signaling. Aside from these well-established roles, NO exerts pleiotropic functions in many different types of cells [20]. These functions include regulation of tissue morphogenesis, polarity formation, cellular growth, and movement [21,22,23,24,25,26].

Because of its nature as a radical, NO (•NO) is also involved in both anti- and pro-oxidant mechanisms. As an anti-oxidant, NO could modify cellular processes to confer protection to cells and tissues against oxidative damage [8]. It is reported that pretreatment with NO could protect cells against the oxidative stress by hydrogen peroxide (H_2_O_2_) [27,28], while NO could also limit the production of ROS by nicotinamide adenine dinucleotide phosphate (NADPH) oxidase in immune cells [29]. In addition, the beneficial effects of pretreatment with NO against ischemia–reperfusion-mediated tissue injury have been reported in clinical trials [30]. Conversely, as a pro-oxidant, NO could react with molecular oxygen or ROS (especially, superoxide: O_2_•^−^) and be converted to a strong oxidant, reactive nitrogen oxide species (RNOS, especially, peroxynitrite: ONOO^−^), that could oxidatively damage a variety of biological molecules.

It is proposed that whether NO exerts anti- or pro-oxidant effects depends on the concentration of NO [8]. At lower NO concentrations, the activities of NO manifest through its direct interaction with the biological target, which are likely to lead to anti-oxidant effects. At higher NO concentrations, conversely, the activities of NO are indirectly mediated by RNOS derived from the reaction of NO with molecular oxygen or ROS, which are likely to lead to pro-oxidant effects. The deleterious effects of NO as a pro-oxidant often take place in confined locales where local NO concentrations could be quickly raised to higher levels [8]. In fact, dose-dependent activities of NO have been reported by a number of previous studies. These studies suggest that the threshold concentration of NO that determines whether the effect is direct or indirect would be around 1 μM [8,31].

Local NO concentration is in part controlled by spatial regulation and compartmentalization of the NO source. There are three isoforms of NOS (NOS1-3) that produce NO in mammalian cells. NOS-1 (neuronal NOS, nNOS) and NOS-3 (endothelial NOS, eNOS) are the constitutive NOS and are part of the membrane-bound protein complexes found in different subcellular locales. NO produced by NOS-1 or NOS-3 quickly (<0.1 sec.) and directly reacts with the proximal targets, including NOS itself and the interacting proteins [3,4,6]. In contrast, NOS-2 (iNOS) is the inducible NOS. In inflammatory cells, NOS-2 is initially expressed in the cytosol, but is recruited to phagosomes or peroxisomes to elevate the local NO concentration, where NO reacts with superoxide, forming peroxynitrite to kill pathogens [32,33]. Thus, dichotomous effects of NO as both an anti- and pro-oxidant are dependent on the source and location of NO that determines its local concentration.

## 3. NO Production and Biochemistry

### 3.1. Nitric Oxide Synthase (NOS) 

Mammalian NOS consists of three isoforms (NOS1-3) that encompass ~50% homology [7]. NOS-1 (nNOS) and-3(eNOS) are expressed constitutively, while their activities are regulated post-translationally, such as by phosphorylation, S-nitrosylation, protein interaction, cofactor/substrate, and calcium level. NOS-1 and NOS-3 produce a steady-state NO level to regulate tissue homeostasis and development [10,34]. Conversely, the expression of NOS-2 (iNOS) is regulated inducibly to produce a large amount of NO in response to an inflammatory signal [10,34]. In addition, mitochondria are reported to possess mtNOS (nNOS homologue) in the matrix and inner membrane. mtNOS is involved in the regulation of oxygen consumption and biogenesis of mitochondria [35,36,37,38,39]. However, the actual presence of mtNOS is currently under debate (see below for details).

NOS is a dimer of two identical monomers tethered by tetrahedral coordination of a zinc ion at two CysXXXXCys motifs (each motif is contributed by a monomer). Through the motif, NOS binds the substrate L-arginine and cofactor tetrahydrobiopterin (BH_4_) which facilitates dimerization, substrate binding, and enzymatic function [34,40]. The constitutive NOS-1 and -3, but not inducible NOS-2, are bound by calmodulin in response to increased calcium levels, which induces conformational changes to activate the enzymatic function [34,41]. Each NOS monomer is composed of two distinct domains: the carboxyl-terminal reductase and amino-terminal oxygenase domains. Both domains are connected by a linker region that binds calmodulin [34,41]. The reductase domain harbors a series of redox-active cofactors, NADPH, flavin adenine dinucleotide (FAD), and flavin mononucleotide (FMN), which serially transfers electrons. The oxygenase domain, serving as the dimerization site that coordinates a zinc ion, harbors another redox-active cofactor BH_4_, heme, and the substrate L-arginine. In particular, BH_4_ plays an essential role in “coupling” the reductase and oxygenase domains [7,34,42,43]. It facilitates electron transfer from FMN of the reductase domain to the heme of the oxygenase domain. This causes the ferric (Fe^3+^) center of the heme to be reduced to the ferrous (Fe^2+^) center, which then reacts with molecular oxygen to form Fe^2+^-O_2_ complex. This complex, in turn, oxidizes the guanidine moiety of the substrate L-arginine, producing L-citrulline and NO [7,34,42,43] (Figure 1, left).

If the availability of the cofactor BH_4_ or the substrate L-arginine is reduced, however, NOS becomes “uncoupled” and unable to dimerize. The uncoupled NOS fails to transfer electrons from the reductase to the oxygenase domains and to oxidize L-arginine. Then, instead of producing NO, the heme of the oxygenase domain now produces superoxide (O_2_•^−^) (Figure 1, right), which further lowers the BH_4_ level by oxidizing it to dihydrobiopterin (BH_2_) [43,44]. In fact, deficiency of BH_4_ is a major cause of NO deficiency in chronic disorders, such as diabetes [45], obesity [46], cardiovascular disease [40,44], as well as cancer [47]. NO deficiency could lead to tissue fibrosis and stiffening [48], which also increases the cancer risks of patients [49].

BH_4_ deficiency could be attributed to its oxidative degradation to BH_2_ under hyperoxia or by a variety of biological oxidants, such as H_2_O_2_, peroxynitrite and heme [50,51]. On the other hand, BH_4_ deficiency could be ascribed to the deficiency of the enzymes that generate BH_4_ in the body. BH_4_ is either *de novo* synthesized from GTP or regenerated from the oxidized metabolites of BH_4_ (i.e., BH_2_ and sepiapterin) [52,53]. A defect of any enzyme involved in the two pathways could lead to BH_4_ deficiency. In particular, a mutation of GTP cyclohydrolase 1 (GCH1)*,* the rate-limiting enzyme of BH_4_ biosynthesis, is a major genetic cause of BH_4_ deficiency found in different types of neurological and metabolic disorders [54,55].

### 3.2. NO Signaling 

NO signaling could be classified into the classical and non-classical schemes. In the classical scheme, NO binds to the heme group of soluble guanylyl cyclase (sGC) to promote the enzymatic activity that converts GTP to cGMP. cGMP serves as a second messenger and activates cGMP-dependent protein kinase (PKG). This lowers the levels of potassium and calcium ions in the cytosol, which hyperpolarizes the membrane potential and triggers neurotransmission (in neurons) and vasodilation (in vessels) [10,34]. Such a classical NO pathway is said to be a “long-range” form of signaling, where the signal is passed through a relatively long distance from the NO sources. An example of this pathway includes paracrine/autocrine signaling of NO [56,57].

The non-classical scheme of NO signaling includes covalent post-translational modifications of biomolecules by NO and NO derivatives. These are S-nitrosylation of protein thiols; metal nitrosylation of transition metals; and oxidative nitration or hydroxylation of various molecules (e.g., tyrosines, thiols, amines, fatty acids, and guanine. See below for further details). This mode of NO pathway is said to be a “short-range” one, since the effect takes place in a relatively close range from the NO source and often within a certain subcellular localization [56]. Nevertheless, recent evidence unveils that protein S-nitrosylation could be propagated beyond the boundary between different subcellular locales through transnitrosylation (see below) [58]. S-nitrosylation is an emerging paradigm of redox signaling by which cells protect themselves against oxidative stress and reduce the levels of ROS [15,59,60,61,62]. We will focus on these antioxidant effects of protein S-nitrosylation in the following sections.

### 3.3. S-Nitrosylation, Metal Nitrosylation and Nitration 

Nitrosylation is the reaction that covalently incorporates the nitrosyl moiety of NO into another molecule. If nitrosylation takes place at the thiol group of a cysteine, this reaction is termed S-nitrosylation. On the other hand, if nitrosylation takes place at a transition metal (e.g., the catalytic site of a metalloenzyme), this reaction is termed metal nitrosylation.

S-nitrosylation takes place on cysteine thiols at a physiological pH range [63]. Although this reaction was discovered over 150 years ago [64], it has only been two decades since S-nitrosylation was recognized as an NO-mediated protein modification [65]. Cumulative evidence has supported that S-nitrosylation is a ubiquitous regulatory mechanism for protein conformational change, protein-protein interactions and further post-translational modifications such as phosphorylation, acetylation, ubiquitination and disulfide bond formation [11,12,13,14,58]. S-nitrosylation regulates diverse processes, including transcription regulation, DNA damage repair, cellular growth/differentiation, and apoptosis [11,18,19]. Balanced regulation of S-nitrosylation is essential for normal pathophysiology, whereas its dysregulation leads to disease conditions [66]. We will discuss the biological activities of S-nitrosylation in detail below.

For metal nitrosylation, NO interacts with metal centers of the heme, as first discovered by Keilin and Hartree 80 years ago [67]. NO binding to the ferrous (Fe II) heme of sGC induces conformational change leading to the activation [68], whereas NO binding to the heme of cytochrome c oxidase in the mitochondrial electron transport chain causes inactivation [69]. NO also binds to the ferrous hemoglobin with the higher affinity than those of oxygen and carbon oxide. In hyperoxic conditions such as ischemia reperfusion, NO binding to hemoglobin reduces the affinity of oxygen binding and confers protection to tissues against oxygen toxicity [70].

In the presence of high concentrations of NO and ROS, in particular superoxide, both molecules react to produce a highly destructive peroxynitrite, which could peroxidize lipids, nitrate tyrosines, thiols, amines and fatty acids, and hydroxylate guanine nucleotides at acidic pH [63]. In particular, tyrosine-nitrated proteins are indicators of oxidative/nitrosative stress [63] and are degraded by 20S proteasome [71]. Nitrated unsaturated fatty acids, on the other hand, form nitroalkenes or nitrohydroxyl derivatives that exert anti-inflammatory signals [72].

## 4. S-Nitrosylation 

### 4.1. Biochemistry of S-Nitrosylation

NO itself is not an oxidant and does not strongly react with a protein thiol. Thus, for the majority of S-nitrosylation, NO first reacts with oxygen to increase the oxidation state and then reacts with a thiol [73]. Different intermediate reactions have been proposed [56].

Case (1) NO reacts with O_2_ to form a series of nitrogen oxides with increasing oxidation states (auto-oxidation). Then, N_2_O_3_ reacts with a protein thiol to produce nitrite and a nitrosothiol. In particular, the rate of this reaction increases in hydrophobic environments such as membranes, the predominant locales of NOS 1-3 (Figure 2, reaction 1) [56].

Case (2) First, NO reacts with O_2_ to form NO_2_, and NO_2_ reacts with a thiol to produce a thiol radical and nitrite. Then, NO reacts with a thiol radical to form a nitrosothiol (Figure 2, reaction 2) [56].

Case (3) In the presence of a thiol radical, NO directly reacts with it to form a nitrosothiol (Figure 2, reaction 3) [56].

Case (4) For a metal-catalyzed reaction, NO becomes oxidized by a transition metal, such as Fe^3+^ or Cu^2+^, in a metalloenzyme, forming nitrosonium (NO^+^). Nitrosonium then reacts with a thiol close to the catalytic center to form a nitrosothiol. This reaction takes place in specific cases, such as auto-nitrosylation of hemoglobin and GSNO formation by cytochrome c (Figure 2, reaction 4) [56].

### 4.2. S-Nitrosylation Reaction Specificity

Despite such abundance, S-nitrosylation occurs only at select cysteine residues of proteins. Such selectivity depends on the following criteria [5]:

***(1) The target cysteines must be in the proximity of the NO source to increase the likelihood of S-nitrosylation*** [5].

(i) The first S-nitrosylation targets are the constitutive NOS (NOS-1 and NOS-3) themselves [3,74]. S-nitrosylation of NOS-3 inhibits its dimerization and thus activation [74] suggesting that S-nitrosylation of the constitutive NOS may serve as a self-shut-off mechanism. Conversely, S-nitrosylation of the inducible NOS, NOS-2, is currently unknown.

(ii) The second S-nitrosylation targets are proteins directly interacting with NOS. NOS-1 interacts with a scaffolding protein DLG4 (a.k.a., PSD95) and a small G-protein Dexras1, which both become S-nitrosylated [3,75]. While the consequence of DLG4 S-nitrosylation is currently unknown, Dexras1 S-nitrosylation activates the protein function that promotes growth suppression and apoptosis of cells [75]. NOS-3 interacts with the Hsp90 chaperon protein, which also becomes S-nitrosylated. Hsp90 S-nitrosylation inhibits its ATPase activity [4] and impairs the growth/survival of cancer cells dependent on Hsp90 chaperon functions [76]. NOS-2 interacts with calcium- and zinc-binding proteins S100A8 and S100A9, which both become S-nitrosylated. S100A8/A9 are pro-inflammatory factors by their very nature. However, upon S-nitrosylation, they become converted to anti-inflammatory agents that inhibit mast cell activation and leukocyte-endothelium interaction (see the details below) [77,78,79].

(iii) The third S-nitrosylation targets are proteins that are transnitrosylated by S-nitrosylated proteins (S-nitrosylases). For example, S100A8/A9, after being S-nitrosylated by NOS-2 [77,78], transfer the nitrosyl group to their interacting proteins such as vimentin [78,80,81], conferring anti-inflammatory effects [79]. See below for more details on this phenomenon.

**(2) *The target cysteine must be within a signature motif, I/L-X-C-X2-D/E, which is specifically recognized by NOS* [78]**. This motif must be within α-helix forming a large surface area to increase accessibility to reactants [78]. Furthermore, within the S-nitrosylation motif, the target cysteine thiol must electrostatically interact with the neighboring charged residues (<6 Å), which increases its nucleophilicity (i.e., reactivity), thereby promoting its reactivity with the nitrosyl group [82,83].

**(3) *The target cysteine must be within a highly hydrophobic region formed by tertiary protein structure or membranes*** [11,84]. Because NO itself has low reduction potential (low tendency to receive electrons; oxidation state: +2), it cannot oxidize amino acids to form adducts at a physiological rate. Thus, NO-dependent amino acid oxidation is preceded by the oxidation of NO by molecular oxygen. This leads to formation of nitrogen oxide [NO_2_ (oxidation state: +4)] and N_2_O_3_ (Oxidation State: +3), which then elevates the reduction potential of the nitrosyl group by up to 10 fold [85,86]. Because NO’s reaction with molecular oxygen is so critical, S-nitrosylation of proteins preferentially takes place at cysteines in hydrophobic regions which could attract hydrophobic gases, NO and molecular oxygen, and enhance their reaction rate by 30–300 fold [84].

**(4) *The target cysteine must be in the suitable environment*.** Which thiols are S-nitrosylated and what is the stability of the nitrosothiols, depend on the context and environmental factors [63]. The accessibility of these target cysteines, which are often masked within hydrophobic regions of the protein, largely depends on the nature of neighboring residues. First, the pKa of the thiol is greatly influenced by the acidity and basicity of neighboring residues. Second, the presence of bulky amino acid residues (e.g., Phe, Tyr, Arg, and Leu) in close proximity elicits steric hindrance onto the target cysteines. Thus, cysteines targeted for S-nitrosylation are those that have lower pKa, are surrounded by acidic and basic residues within 6 Å (promoting nucleophilicity of thiols), and are adjacent to few bulky residues (less sterically hindered) within 8 Å [87]. Furthermore, at an acidic pH and in the presence of ROS, the same thiols could instead be targeted for nitration (oxidation) by NO as discussed above [63], which would then counteract their S-nitrosylation [15].

### 4.3. Other Factors that Control S-Nitrosylation Level

The level of S-nitrosylation in the cell is also regulated by the following factors:

**(1) *Redox status.*** Redox status modulates *S*-nitrosylation. High levels of antioxidants, which elevate the reducing potential of cells, could prevent S-nitrosylation, whereas a decrease of antioxidants promotes S-nitrosylation [61,88,89]. For example, depletion of an antioxidant, glutathione (GSH), elevates S-nitrosylation of mitochondrial proteins (see below), which protects cells against thiol oxidation, permeabilization and apoptosis [90].

**(2) *Denitrosylation.*** S-nitrosylation levels in cells are controlled by the balance between S-nitrosylation and denitrosylation. While S-nitrosylation is generally a non-enzymatic reaction (except for prokaryotes), denitrosylation can be a non-enzymatic or enzymatic reaction [91]. Cleavage of S-nitrosyl group could spontaneously occur in the presence of reducing agents (ascorbate, glutathione), metal ions (Cu^2+^), heat, UV, ROS, or nucleophiles (or anions) [91]. In contrast, denitrosylation could be catalyzed by denitrosylases that enzymatically remove the S-nitrosyl group from cysteines. There are two major denitrosylase systems in cells: Thioredoxin (Trx)/thioredoxin reductase (TrxR) and S-nitrosoglutathione (GSNO)/GSNO reductase (GSNOR) systems. Trx/TrxR targets S-nitrosylated proteins that harbor C-X_5_-K or C-X_6_-K motif, including caspase-3 and nuclear factor-κB (NF-κB) [91]. On the other hand, GSNOR is the alcohol dehydrogenase 5 (ADH5), and GSNO (the major endogenous S-nitrosyl donor, see below) is the only substrate of denitrosylation. GSNOR activity is inhibited by ROS, such as hydrogen peroxide, which elevates GSNO and antioxidant gene expression [92]. The balance between S-nitrosylation and denitrosylation is essential for normal physiological conditions. Overexpression or defect of GSNOR dysregulates S-nitrosylation levels and could trigger many diseases, including multi-organ dysfunction, sepsis and cancer [91]. For example, the increase of GSNOR level in the aging brain contributes to cognitive impairment [93].

**(3) *Endogenous NOS inhibitors*.** S-nitrosylation level could be regulated by NOS inhibitors that modulate NO levels. There are several endogenous NOS inhibitors which are classified into two categories: (1) L-arginine analogues (competitive inhibitors) and (2) allosteric NOS inhibitor.

(1) L-arginine analogues: endogenous methylarginines (MAs): There are three endogenously produced MAs: asymmetric dimethylarginine (ADMA); symmetric dimethylarginine (SDMA) and N^G^-monomethyl-L-arginine (L-NMMA). They are formed by the liberation of methylated arginine residues from intracellular proteins. Out of the three MAs, ADMA and L-NMMA could bind the three isoforms of NOS at the L-arginine binding site to exert competitive inhibition. Both MAs are found in the blood, where ADMA is 5–10 times more abundant than L-NMMA [94]. In particular, ADMA levels are elevated in patients with different diseases such as cardiovascular disease, diabetes, Alzheimer’s disease, and liver failure and kidney failure [94].

(2) Allosteric NOS inhibitor: Dynein light chain LC8-Type 1 (DYNLL1, PIN) is the 8 kDa cross-linker of dynein to cargos and to adapter proteins. DYNLL1 binds to the N-terminal PDZ domain of NOS1 (absent in NOS2 and NOS3) and inhibits its dimerization, inactivating the enzymatic function. DYLL1 level quickly increases in response to ischemia, which is intended to counteract the rise in NOS1 activity to protect the tissue against damage by excessive NO [95].

## 5. Transnitrosylation

### 5.1. Reaction

The selectivity of S-nitrosylation sites may be ameliorated by successive S-nitrosylation reactions termed transnitrosylation [58]. Transnitrosylation is a process where a protein (nitrosylase), which is S-nitrosylated at cysteine(s) or nitrosylated at the metal center (e.g., heme), interacts with a protein containing the I/L-X-C-X2-D/E motif, leading to the transfer of the nitrosyl moiety to the interacting cysteine thiol. Such nitrosylation may take place successively, allowing for signal transmission to a site distant from the NO source. These reactions take place in two scenarios: Cys-to-Cys or Metal-to-Cys transnitrosylation (Figure 3) [58,78,85].

In the metal-mediated transnitrosylation, the nitrosyl group may be transferred intramolecularly or intermolecularly. In hemoglobin, NO is transferred from the heme iron to the neighboring cysteine thiols in the same molecule, whereas in cytochrome c, the metal-coordinated NO is transferred to the thiols of an interacting glutathione leading to the formation of GSNO, which is discussed further below [66,96].

Transnitrosylation involves the nucleophilic attack of the recipient’s thiolate anion on the nitrosyl nitrogen of the donor (nitrosylase) [85]. So far, less than 10 S-nitrosylases have been identified, and only specific cysteines are targeted for S-nitrosylation. This enables selective activation/inhibition of particular signaling pathways [58]. A major determinant for the selective NO transfer is the physical distance between the donor (S-nitrosylase) and recipient thiol groups of cysteines. Another determinant is the redox potential between the donor and recipient thiols [58]. Thus, transnitrosylation occurs only when two proteins directly interact and possess appropriate redox potentials allowing for electron transfer followed by NO transfer [58]. It is speculated that physical association of the two proteins triggers conformational change, allowing the recipient thiol to form thiolate anion (RS^−^) which then attacks the nitrosyl group of the donor [58]. There are fewer than 10 transnitrosylation reactions that are known to date [58]. Among these reactions, some of the major signaling pathways are listed below.

### 5.2. S-Nitrosylases or Transnitrosylases

#### 5.2.1. S-Nitrosoglutathione (GSNO)

S-nitrosoglutathione (GSNO) is the most abundant S-nitrosothiol and the major endogenous NO donor for proteins everywhere in the cell. GSNO is generated in the mitochondria when the nitrosyl group is transferred from the heme iron of cytochrome c to glutathione (GSH) [96]. GSNO then translocates to different subcellular locales and transnitrosylates the interacting proteins including NF-κB, STAT3, AKT, EGFR, and IGF-1R [97,98,99]. GSNO-mediated transnitrosylation of the p65 and p50 subunits of NF-κB, as well as IκB kinase, inhibits NF-κB activation that is closely associated with malignant behavior of cancers [100,101,102]. One of the NF-κB-mediated oncogenic signaling is to upregulate IL6, which then activates STAT3, a transcription factor that promotes cell survival and proliferation [103]. STAT3, downstream of NF-κB, is also targeted for transnitrosylation by GSNO, which inhibits STAT3 phosphorylation required for its activation [101]. Additionally, cell surface receptors and the associated proteins, such as AKT, EGFR, and IGF-1R, are transnitrosylated by GSNO, leading to inhibition of their phosphorylation-dependent activation [99]. Accordingly, preclinical studies have reported the potent anti-cancer effects of GSNO in suppressing tumor cell growth and improving the efficacy of radiation therapy [98,101].

#### 5.2.2. Glyceraldehyde-3-Phosphate Dehydrogenase (GAPDH)

Another important S-nitrosylase is the glycolytic enzyme glyceraldehyde-3-phosphate dehydrogenase (GAPDH). GAPDH is well known for its role in catalyzing the conversion of glyceraldehyde 3-phosphate to D-glycerate 1,3-bisphosphate in the cytosolic glycolysis [104]. GAPDH also plays critical roles as a transnitrosylase in the nucleus and mitochondria, including transcriptional regulation and apoptosis [105].

GAPDH translocates from the cytosol to the nucleus upon S-nitrosylation at the catalytic Cys150 by S100A8/A9 that has been S-nitrosylated by NOS-2 in response to a stress [78,106]. S-nitrosylation of GAPDH enables its interaction with an E3 ubiquitin ligase Siah1 which has the nuclear localization signal (NLS). GAPDH-Siah1 complex then translocates to the nucleus, where Siah1 mediates ubiquitination and degradation of nuclear proteins to initiate apoptosis. In contrast, GAPDH forms a complex with p53 to activate p53-mediated apoptosis [105,107,108,109]. Furthermore, GAPDH transnitrosylates proteins involved in transcription and DNA repair. These proteins include deacetylases: sirtuin-1 (SIRT1) and HDAC2, inhibited by S-nitrosylation; and DNA repair protein: DNA-activating protein kinase (DNA-PK), activated by S-nitrosylation [105,106].

In response to a stress, GAPDH also translocates to mitochondria to transnitrosylate mitochondrial proteins, such as Hsp60, acetyl-CoA acetyltransferase (ACAT1) and voltage-dependent anion channel 1 (VDAC1) [110]. The elevation of GAPDH-mediated transnitrosylation of mitochondrial proteins regulates the permeability of the mitochondrial membrane, mitochondrial functions and cell death [111].

In cancer cells, GAPDH-mediated transnitrosylation of nuclear proteins is impaired, leading to a defect of stress-responsive apoptosis [106,108,112,113]. One reason for this impairment is the downregulation of Siah1 that facilitates the nuclear translocation of GAPDH [112]. Siah1 expression is directly regulated by p53, the tumor suppressor downregulated in many cancer types [112,114].

#### 5.2.3. S100A8/S100A9

The third example of S-nitrosylases are S100A8 and S100A9, which are calcium- and zinc-binding proteins playing roles in the regulation of inflammatory processes and immune response, such as neutrophil chemotaxis and adhesion [77]. They represent up to 40% of neutrophil cytoplasmic proteins, but are secreted in response to the elevated level of ROS at the site of inflammation [77,115]. S100A8 and S100A9 predominantly exist as calprotectin (S100A8/A9 heterodimer) which forms a complex with arachidonic acid (AA) released upon tissue injury. S100A8/A9-AA complex interacts with NADPH oxidase (NOX), and AA is transferred to NOX to activate its catalytic activity to produce ROS, as is discussed further below [116].

However, S100A8/A9 are both targeted for S-nitrosylation and converted to anti-inflammatory agents that inhibit mast cell activation and leukocyte-endothelium interaction [79,115]. Furthermore, S-nitrosylated S100A8/A9 transnitrosylates their interacting proteins to transmit anti-inflammatory signals to distant sites. Upon NOS-2 induction under inflammatory stimuli, S100A8/A9 assists in the formation of a linkage between NOS-2 and the target proteins, allowing the nitrosyl group to be transferred from NOS-2 to the target [77,78]. Over 100 proteins transnitrosylated by S100A8/A9 have been so far identified within cells and in microcirculation [78]. These targets include GAPDH and hemoglobin (see above). S100A8/A9 also transnitrosylates some cytoskeletal elements such as ERM proteins (ezrin and moesin), linking cortical actin to the plasma membrane, and vimentin, the major intermediate filament in mesenchymal cells and metastatic cancer cells [78,79,80,81]. The direct consequence of S-nitrosylation of these cytoskeletal elements is yet to be determined. Nevertheless, it is proposed that S-nitrosylation of these proteins induces conformational changes, influencing protein stability and protein-protein interactions [117,118].

## 6. S-Nitrosylation of Cellular Proteins for Redox Regulation 

Inside the cell, S-nitrosylation levels of cytosolic proteins are relatively low. One reason for this is that the S-nitrosyl group could be spontaneously cleaved by various factors in the cytosol. These factors include reducing agents (ascorbate and glutathione), metal ions (Cu^2+^), heat, UV, ROS, and nucleophiles (or anions) [17,91]. The other reason is that the subcellular localizations of the constitutive NOS-1 and NOS-2 are compartmentalized, which also compartmentalizes the majority of S-nitrosylation reactions [3,4,6,117]. Although NOS-2 is initially expressed in the cytosol, NO produced by NOS-2 is more likely to react with ROS to form RNOS than S-nitrosylating protein thiols [32,33]. Thus, the major sites of S-nitrosylation are organelles, (e.g., mitochondria, Golgi apparatus, and endoplasmic reticulum), nucleus, and plasma membrane. Among these sites, we will feature mitochondria and nuclei where S-nitrosylation of proteins plays critical roles in protection of cells against oxidative stress. Some of these proteins are listed in Table 1.

### 6.1. S-Nitrosylation of Redox-Sensitive Mitochondrial Proteins

Mitochondria play key roles in the production of ROS as well as conferring antioxidant mechanisms along with all other functions (Figure 4) [134]. The bioactivities and quality control of mitochondria are in part regulated by S-nitrosylation and denitrosylation of mitochondrial proteins [135,136]. Mitochondrial proteins become S-nitrosylated in response to changes in mitochondrial respiration and redox equilibrium. This results in the protection of protein thiols and the cell against oxidative damage and death, while also preventing further ROS production [59,60,61]. Such responsiveness of S-nitrosylation to the oxidizing environment could be in part ascribed to the nature of its biochemistry. NO itself is not an oxidant and would need to react with molecular oxygen to increase the oxidation state in order to react with a protein thiol [73].

The source of mitochondrial NO remains under debate. On one hand, mitochondrial NO may be generated by mitochondrial NOS (mtNOS) [60]. mtNOS is reported to be calcium-sensitive and constitutively active (homologous to NOS-1 [nNOS]) and integral to the inner mitochondrial membrane [36]. However, the existence of mtNOS remains controversial [60]. On the other hand, mitochondrial NO may be derived from NO generated in the cytoplasm and transported via a transnitrosylase through a cognate transporter on the mitochondrial membranes [61,137].

NO produced in the mitochondria could S-nitrosylate or Fe-nitrosylate proteins preferentially in the inner mitochondrial membrane and intermembrane space because of their lower pH and lipophilic environments. Mitochondrial proteins targeted for S-nitrosylation include all the complexes in the electron transport chain (ETC): Complex I (NADH: ubiquinone oxidoreductase); Complex II (succinate dehydrogenase); Complex III (cytochrome b-c1 Complex); cytochrome c, Complex IV (cytochrome C oxidase) as well as adenosine triphosphate (ATP) synthase (Complex V) [59,60]. In addition, several metabolic enzymes involved in citric acid (Krebs) cycle and β-oxidation [60,138], as well as different mitochondrial proteins involved in regulation of apoptosis [119,139,140], are S-nitrosylated. Such S-nitrosylation of mitochondrial proteins mostly inhibit their activities to regulate mitochondrial O_2_ consumption, ROS production, and mitophagy, while protecting the cell against death signals (Figure 4) [36].

On the other hand, in the presence of large amounts of ROS (and/or NO), NO could react with superoxide to produce a powerful oxidant, peroxynitrite (ONOO^−^) that can irreversibly nitrate or peroxidate tyrosines and unsaturated fatty acids [137,141].

#### 6.1.1. Electron Transport Chain (ETC) and ATP Synthase 

The electron transport chain (ETC, respiratory chain) is composed of Complexes I-IV bound to the inner membrane of the mitochondria. Electrons are sequentially passed from Complex I to IV through a series of redox reactions, which releases energy to generate proton gradient across the inner membrane. Proton gradient then triggers chemiosmosis to drive ATP synthesis (Complex V). The entire process, termed oxidative phosphorylation, is the major source of ATP in aerobic organisms [142]. On the other hand, ETC is the primary site of ROS production in the cell. ROS production becomes the most prominent when ETC and ATP synthesis are uncoupled due to the insufficient proton gradient across the inner membrane. Such uncoupling causes reverse electron transport (RET) at Complex I, leading to production of large amounts of superoxide [143]. The amount of ROS generated may often exceed the amount catalyzed by mitochondrial antioxidant enzymes, superoxide dismutase (SDS) and catalase (CAT).

In oxidizing conditions, the enzymatic activities of Complexes I-V could be inhibited by S-nitrosylation or Fe-nitrosylation, preventing further ROS production [59,60,61]. For example, O_2_ binding to the Fe-heme center of Complex IV could be competitively inhibited by NO binding, which reduces oxygen consumption and thus further the production of ROS [144]. In addition, S-nitrosylation of Complexes not only protects themselves against oxidative damage by ROS, but also scavenges NO to prevent it from reacting with superoxide to form peroxynitrite [141]. Furthermore, in hypoxic conditions, nitric oxide production becomes elevated [145], which could also promote S-nitrosylation and inactivation of mitochondrial ETC proteins [144]. This reduces mitochondrial oxygen consumption and allows for the redistribution of intracellular oxygen to other locales. This would also prevent the stabilization of hypoxia-inducible factor-1α (HIF-1α) in response to hypoxia [146].

Inactivation of Complexes I-V ultimately impedes oxidative phosphorylation (i.e., cellular respiration), which disrupts protein pumping and causes mitochondrial depolarization [39,59]. This activates the PINK1/Parkin pathway that induces mitophagy to selectively eliminate dysfunctional mitochondria [147]. Mitophagy is a mechanism by which mitochondria protect the cell against oxidative stress [148].

In addition to the five Complexes, cytochrome c, which carries electrons between Complexes III and IV, is also targeted for S-nitrosylation. Cytochrome c becomes nitrosylated at the iron of the heme moiety, which then transfers the nitrosyl group to glutathione (GSH), generating the abundant transnitrosylase, GSNO [149]. Although S-nitrosylated cytochrome c is released to the cytosol during apoptosis [150], it retains its ability to synthesize GSNO and is proposed to exert suppression on apoptotic signaling [149].

#### 6.1.2. Metabolic Enzymes 

Mitochondrial inner membrane and matrix are the sites of aerobic oxidation of pyruvate (citric acid [Krebs, TCA] cycle) and fatty acids (fatty acid β-oxidation) to CO_2_. These catabolic reactions could not only generate ROS by themselves [151,152,153], but also serve as the sources of electrons and co-enzymes (e.g., NADH and FADH_2_) to drive the ETC that produces ROS [152,154]. Under oxidizing conditions, many of these catabolic enzymes are targeted for S-nitrosylation to inhibit their activities [59,60,138,155]. These include three enzymes from citric acid (Krebs) cycle (aconitase; *α*-ketoglutarate dehydrogenase; and the succinate dehydrogenase [Complex II]); and five from fatty acid β-oxidation (long-chain and short-chain acyl-CoA dehydrogenase; enoyl-CoA hydratase; carnitine palmitoyl transferase 2 and flavoprotein dehydrogenase) [60]. Inhibition of these catabolic enzymes contributes to the reduction of ROS production.

#### 6.1.3. Mitochondrial Pro- and Anti-Apoptotic proteins 

As described above, S-nitrosylation of the ETC and mitochondrial metabolic enzymes in the oxidizing environment prevents further production of ROS and protects protein thiols against oxidative damage. On the other hand, S-nitrosylation of pro- and anti-apoptotic/necrotic mitochondrial proteins protects the cell against death. The reason for such dichotomous effects could be that because reduced energy production (by inhibition of the ETC and metabolic enzymes) would otherwise induce cell death, death pathways are also inhibited for counteraction. These pro- and anti-apoptotic proteins include mitochondrial permeability transition pore (mPTP), voltage-dependent anion channel (VDAC), caspases, and Bcl-2.

mPTP is formed in the inner mitochondrial membrane under a stress (e.g., oxidative stress) and plays a pivotal role in controlling the mitochondrial membrane potential. Opening of the pore under stress increases the permeability of the mitochondrial membrane to molecules <1.5 kDa, leading to mitochondrial swelling and necrosis [156]. NO produced at the basal level (e.g., 5 μM) could S-nitrosylate cyclophilin D (CypD), a critical mPTP regulatory component. This prevents the association of CypD with mPTP that is required for opening the pore [139,157] and confers a protection to the cell under a stress. On the other hand, NO produced at a high concentration (e.g., 500 μM) could produce peroxynitrite in the presence of large amounts of ROS. Peroxynitrite could oxidize mPTP to induce multiple disulfide-linkages, which would lead to the opening of mPTP, loss of ATP production, and necrosis [158].

VDAC is located in the mitochondrial outer membrane and controls the influx and efflux of mitochondrial metabolites. In addition, VDAC plays a major role in mitochondrial-mediated apoptosis through regulation of the release of cytochrome c that activates caspase-9 for apoptotic signaling [159]. The effect of S-nitrosylation on VDAC is reported to be biphasic, similar to mPTP. Low levels (<1 μM) of NO (and S-nitrosylation) inhibit VDAC function, conferring protection to the cell against apoptotic activation. On the other hand, higher levels of NO upregulate the function [160], although it is unknown whether this also involves the production of peroxynitrite or other mechanisms.

Caspases are the major executors of apoptosis located in the cytoplasm and mitochondrial intermembrane space. In the absence of apoptotic signal, zymogens of caspase-3 and caspase-9 in mitochondria are S-nitrosylated at the catalytic sites to prevent their activities. Upon activation of Fas receptor, zymogens become denitrosylated, allowing for their further activation [119].

Bcl-2 family proteins are localized on the outer mitochondrial membrane and regulate apoptosis by controlling the release of pro-apoptotic intermembrane proteins (e.g., cytochrome c) [161]. The anti-apoptotic Bcl-2 becomes S-nitrosylated in response to apoptotic stimuli. This inhibits ubiquitin-proteasomal degradation of Bcl-2 and confers protection to the cell against apoptosis [140].

### 6.2. S-Nitrosylation of REDOX-Regulatory Nuclear Proteins

A number of nuclear proteins are targeted for S-nitrosylation in response to a stress, in particular oxidative stress, leading to up- or down-modulation of the activities. For the potential source of the nitrosyl group in the nucleus, two possibilities have been proposed: one is by NOS translocated to the nucleus; and the other is by transnitrosylases localized in the nucleus. NOS 1–3 have been shown to be translocated to the nucleus in response to a stimulus, such as a mitogen, oxidative stress and other pathological conditions. In the nucleus, NOS isoforms regulate S-nitrosylation of nuclear proteins, gene expression and calcium homeostasis [162,163,164,165]. On the other hand, a transnitrosylase, GAPDH, becomes S-nitrosylated in the cytosol in response to a stress (such as oxidative stress) and translocates to the nucleus, where it transnitrosylates a number of nuclear proteins, including transcription factors, nuclear translocators and DNA repair proteins, involved in redox regulation and protection of the cell against stress.

#### 6.2.1. Transcription Factors

Several redox-regulatory transcription factors are S-nitrosylated in response to a change in redox homeostasis, which alters the expression of their target genes to protect the cell against oxidative stress. These transcription factors include nuclear factor (erythroid-derived 2)-like 2 (NRF2); p53, nuclear factor κB (NF-κB); and hypoxia-inducible factor-1α (HIF-1α).

NRF2 is the master regulator of the transcription of antioxidant genes. In response to oxidative stress, NRF2 binds a vast range of gene promoters that harbor the antioxidant-response elements (ARE). These genes include enzymes involved in glutathione and thioredoxin systems (e.g., glutamate-cysteine ligase catalytic subunit [GCLC]), mitochondrial superoxide dismutase (Mn^2+^ SOD, SOD2), peroxiredoxin, HSP70 and ferritin [166,167]. NRF2 is sequestered in the cytoplasm by formation of a latent complex with Klech-like ECH-associated protein 1 (Keap1), whereas dissociation from Keap1 enables nuclear translocation and activation of NRF2 [168]. Among all other stimuli, S-nitrosylation of Keap1 is one of the major mechanisms that induces NRF2 activation [169].

p53 tumor suppressor protein is another transcription factor involved in redox regulation. In response to a mild oxidative stress, p53 becomes S-nitrosylated at Cys124 in the DNA binding domain by the nuclear-localized nNOS. p53 then binds and transactivates the target genes, such as peroxisome proliferator-activated receptor gamma coactivator 1-alpha (PGC-1α), a powerful controller of antioxidant mechanisms. PGC-1α interacts with NRF2 and serves as the co-activator of NRF2-mediated expression of a host of antioxidant genes (e.g., SOD2 and GCLC, see above) [170,171,172]. S-nitrosylation also takes place in MDM2, an E3 ubiquitin protein ligase that binds and targets p53 for degradation. S-nitrosylation of MDM2 inhibits its interaction with p53, thus stabilizing p53 [173]. Although Cys124 of p53 is not a mutation hotspot, nuclear S-nitrosylation of p53 could decline as the nuclear localized nNOS decreases during aging [171].

NF-κB is a transcription factor that regulates the expression of genes critically involved in inflammation and cell proliferation. NF-κB elevates mitochondrial ETC and ROS production, while counteracting the NRF2 pathway [174]. NF-κB is usually bounded by the inhibitor I-κB and sequestered in the cytoplasm. However, in response to a stress such as ROS, I-κB becomes phosphorylated by the I-κB-kinase complex (IKK-α, -β, and -γ), which leads to ubiquitin-proteasome-mediated degradation of I-κB and nuclear translocation and activation NF-κB [175]. IKK- β could be inhibited by S-nitrosylation of the catalytic site, which prevents activation of NF-κB [176] and contributes to the reduction of mitochondrial ROS production [177].

HIF-1 is a transcription regulator that is regulated by oxygen levels. HIF-1 is a heterodimer composed of subunits HIF-1α and HIF-1β. Under normoxic condition, HIF-1α is hydroxylated at proline and arginine residues, which leads to its degradation mediated by the ubiquitin-proteasome system. In response to hypoxia, hydroxylation is inhibited, and the stable HIF-1 dimer translocates to the nucleus, where it transactivates hypoxia-responsive genes, such as vascular endothelial growth factor (VEGF) that promotes angiogenesis [178]. However, during hypoxia, NO could help suppress HIF-1 activation by inhibiting mitochondrial ETC via S-nitrosylation, which then increases the oxygen availability in the intercellular compartments outside mitochondria [14]. On the other hand, in normoxia, NO could S-nitrosylate HIF-1α, which helps stabilize HIF-1α and activate HIF-1-mediated expression of the target genes [179], demonstrating the context-dependence of NO-mediated HIF-1 regulation.

#### 6.2.2. Nuclear Translocator 

The activities of signaling proteins and transcription factors are often regulated by their translocation between the nucleus and cytosol. This process is facilitated by the karyopherin family members that recognize the nuclear localization signal (NLS) or nuclear export signal (NES) of the cargos and translocate them through the nuclear pore complex [180]. One of these transporters is chromosomal region maintenance 1 (CRM1; a.k.a., Exportin-1), which is involved in redox-regulation.

CRM1 facilitates nuclear export of over 200 nuclear proteins that possess the leucin-rich nuclear export signals (NESs) and play pivotal roles in homeostasis and stress response of cells. These cargo proteins include NRF2, p53, p73, MDM2, BRCA1/2, SMAD1/4, STAT1, IκB, c-Abl, and FOXO-3A [181,182,183,184]. These cargo proteins are exported from the nucleus by CRM1 and sequestered or targeted for degradation in the cytosol [184,185]. CRM1 activity is inhibited by S-nitrosylation upon an increase of the nitrosyl group in the nucleus. S-nitrosylation compromises the ability of CRM1 to recognize and bind NESs [184]. One of the major results of such CRM1 inhibition is the nuclear accumulation of NRF2 and induction of the antioxidant defense mechanism, as can be seen above [184].

#### 6.2.3. DNA Damage Repair Proteins

Some DNA damage repair proteins in the nucleus interact with and become transnitrosylated by S-nitrosylated GAPDH. These proteins include the catalytic subunit of DNA-activated protein kinase (DNA-PKcs) [106,186] and Apurinic-apyrimidinic (AP) endonuclease 1 (APE1) [187].

DNA-PKcs play critical roles in the repair of the damaged DNA through non-homologous end joining [188]. DNA-PKcs interacts with and become transnitrosylated by GAPDH which has translocated to the nucleus upon S-nitrosylation [78,106]. Transnitrosylation of DNA-PKcs upregulates its DNA repair activity in response to DNA-damaging anti-tumor agents [122,189].

APE1 (also known as redox factor 1, Ref-1) is involved in the base excision repair of damaged DNA as well as regulation of cellular redox response [190]. Although APE1 is predominantly localized in the nucleus, its subcellular localization is dynamically regulated and it could translocate to the mitochondria and cytoplasm [191]. APE1 interacts with GAPDH which has translocated to the nucleus upon S-nitrosylation [78,106]. APE1 interaction with GAPDH is not only critical for its nuclease activity [187], but also induces transnitrosylation of APE1. Transnitrosylation of APE1 triggers its nuclear export, independent of the nuclear exporter CRM1 (see above), allowing for its translocation to different subcellular locales [192].

## 7. S-Nitrosylation of Extracellular Proteins for Redox Regulators 

The extracellular environment is oxidizing, characterized by the presence of a vast array of proteins linked via disulfide bonds. This is in stark contrast with the intracellular environment that has a highly reducing nature, limiting the formation of disulfide bonds [193]. ROS in the extracellular environment could be derived from that produced in the intracellular environment which passes through a specific transporter (e.g., aquaporin 1, as can be seen below). However, in certain types of cells (e.g., immune cells and vascular cells), NADPH oxidase (NOX) is the major source of the extracellular ROS, producing and secreting ROS into the extracellular environment and circulation (Figure 5) [193]. In addition, protein disulfide isomerase (PDI) complexes with NOX to promote the production of the extracellular ROS. To counteract the extracellular ROS, cells secrete different antioxidant enzymes, including extracellular superoxide dismutase (Ec-SOD, SOD3) which catalyzes the dismutation of superoxide to hydrogen peroxide [193], catalase (CAT), glutathione peroxidase (GPx), and thioredoxin (Trx), which remove hydrogen peroxide [194,195,196]. All these proteins are targeted for S-nitrosylation to regulate their activities.

### 7.1. Sources of the Extracellular NO and Nitrosyl Group

Cumulative evidence unveils that a number of extracellular proteins are S-nitrosylated [197]. This has led to active discussion of the possible sources of the extracellular nitrosyl group (Figure 5). There have been at least three mechanisms proposed: One possible mechanism is that NO passes through the membrane by itself. It has been reported that NO could freely diffuse through the phospholipid bilayer as efficiently as molecular oxygen [198]. In addition, NO could be actively transported through a water channel protein aquaporin-1 (AQP1) [199]. Such NO efflux is proposed to play pivotal roles in regulating NO’s functions in diverse tissues, including immune cells, neurons and vasculature [199,200,201]. The second possible mechanism is that the nitrosyl group is transferred from a major transnitrosylase, GSNO, to a free amino acid cysteine. Then, S-nitrosylated cysteine passes through the cognate amino acid transporter and enters the extracellular space [202]. The third possible mechanism is that secretory proteins become S-nitrosylated prior to secretion. Secretory proteins are processed in the Golgi apparatus prior to exocytosis. It is reported that many proteins found in the Golgi are S-nitrosylated by the Golgi-localized NOS3 [203,204].

### 7.2. Proteins that Regulate Extracellular ROS and NO Levels 

#### 7.2.1. Aquaporin-1 (APQ-1)

APQ-1 is a membrane-bound water channel protein expressed in various types of tissues. It transports water as well as low molecular weight gases such as NO and hydrogen peroxide between the intracellular and extracellular spaces [199,205]. APQ-1 is targeted for S-nitrosylation at the cysteine residue within the functional pore [206,207]. S-nitrosylation is mediated by NO produced from NOS3, co-localized with APQ1 within caveolae. This leads to the inhibition of the channel function of APQ-1, suggesting a possible negative feedback regulation by NO [194].

#### 7.2.2. NADPH Oxidase (NOX) 

NOX is a membrane-bound enzyme complex containing the catalytic heme moiety. NOX is localized on the plasma membrane as well as on the membrane of phagosomes of vascular cells and immune cells. NOX complex faces the extracellular space and produces superoxide or hydrogen peroxide into the extracellular environment by transferring electrons from NADPH to molecular oxygen bound to a heme [208]. NOX is activated by binding arachidonic acid (AA), transferred from S100A8/A9-AA complex, as is discussed above [116]. AA binding facilitates the assembly of the NOX enzyme complex [209]. In disease conditions, NOX2 and NOX4 expression could be elevated by TGF*β* signaling, which could exacerbate oxidative tissue injury [210]. On the other hand, NOX enzymatic activity is inhibited by S-nitrosylation of the cytosolic subunit p47phox, the organizer of the complex, reducing ROS production. Such NO-mediated suppression of NOX is compromised in atherosclerosis, primarily due to the reduced bioavailability of vascular NO [211,212].

#### 7.2.3. Xanthine Oxidase (XO)

Xanthine oxidase (XO) is an enzyme that catalyzes the oxidation of hypoxanthine (deaminated adenine) to xanthine and then xanthine to uric acid for purine catabolism. These reactions also generate superoxide and hydrogen peroxide as byproducts [213]. In response to a stress, such as a high cholesterol level and metabolic changes, XO is released into the extracellular space or circulation and binds to glycosaminoglycans on the target cell surface, increasing the extracellular ROS levels [213,214]. XO-mediated ROS production is inhibited by its direct interaction with NOS1, suggesting the possible involvement of S-nitrosylation [215]. In leptin-deficient obesity syndrome, NOS1 expression is reduced in the cardiac muscle, which overactivates XO to exacerbate oxidative stress, contributing to myocardial dysfunction [216].

#### 7.2.4. Extracellular Superoxide Dismutase (Ec-SOD) 

SOD catalyzes the dismutation of superoxide to peroxide and, in conjunction with catalase which converts peroxide to water, plays a major role in lowering ROS levels. Ec-SOD is a secretory SOD found in most tissues. Once secreted, it is localized in the glycocalyx (glycol-proteins and lipids covering the plasma membrane) of most types of cells, serving as a major extracellular antioxidant enzyme [193]. Ec-SOD also binds to type I collagen in the extracellular matrix. Collagen is composed of 16 % prolines with which ROS could react to cause peptide bond cleavage. Ec-SOD binding confers protection to collagen against oxidative fragmentation [217]. Ec-SOD also plays essential roles in determining the bioavailability of NO by preventing NO from reacting with superoxide to form peroxynitrite [218]. However, Ec-SOD activity itself is influenced by neither NO nor S-nitrosylation [219].

### 7.3. Other Secretory Proteins Regulated by S-nitrosylation for Redox Control

#### 7.3.1. Antioxidant Enzymes

There are three major antioxidant enzymes, other than Ec-SOD, secreted into the extracellular environment to exert redox control. These enzymes are catalase (CAT), glutathione peroxidase (GPx) and thioredoxin (Trx). CAT catalyzes the decomposition of hydrogen peroxide to water and oxygen. GPx reduces hydrogen peroxide to water, while oxidizing glutathione to glutathione disulfide. Trx catalyzes the reduction of oxidized cysteines and removal of hydrogen peroxide. All the enzymes are targeted for S-nitrosylation, which inhibits their enzymatic activities to degrade hydrogen peroxide, possibly as a negative-feedback mechanism [194,195,196]. In fact, the physiological level of hydrogen peroxide is necessary for certain types of redox signaling, whereas its excessive accumulation could confer oxidative stress [220]. For example, the physiological level of hydrogen peroxide inactivates redox-regulated proteases, cathepsin and calcineurin, that degrade the extracellular matrix (ECM) [221,222]. On the other hand, the excessive levels (>100 nM) of hydrogen peroxide could oxidatively damage macromolecules, while sustained (>60 h) production of hydrogen peroxide could activate matrix metalloproteinase (MMP) that degrades the ECM [220,223].

#### 7.3.2. Protein Disulfide Isomerase (PDI)

PDI is an enzyme that catalyzes the formation of disulfide bonds of proteins. PDI is secreted by platelets and endothelial cells during vascular injury to trigger thrombus formation. PDI is also upregulated in response to endoplasmic reticulum stress and disease conditions, such as cancer, serving as a chaperon to correct protein misfolding. Furthermore, PDI associates with NOX to promote the stability of the enzyme complex and the production of the extracellular ROS [224]. PDI is targeted for S-nitrosylation to inhibit its enzymatic activity. S-nitrosylated PDI also serves as a transnitrosylase to propagate the S-nitrosyl signal. The consequence of S-nitrosylation of PDI depends on the context. In the healthy vasculature, where the optimal NO level is maintained, PDI is S-nitrosylated to suppress thrombus formation for the maintenance of vascular quiescence [225]. On the other hand, in neurodegenerative diseases, where high levels of NO are produced, PDI is hyper-S-nitrosylated, which causes the over-accumulation of misfolded proteins and contributes to the disease pathogenesis [226].

## 8. Dysregulated S-Nitrosylation in Disease Pathogenesis 

S-nitrosylation is a key mechanism through which nitric oxide signaling is propagated not only within the cell and tissue, but also in the microenvironment. Under normal physiological conditions, S-nitrosylation of proteins is highly regulated to maintain redox homeostasis. However when encountered with extreme hostile conditions, such as exposure to pathogens, radiation, and carcinogens, this regulation could be interrupted, leading to the occurrence of diseases, most notably, cancer [18]. Recent studies have indicated that S-nitrosylation is a major regulator of tumor microenvironment (TME), the surroundings of cancer cells that critically control the disease pathogenesis [123,128,227]. Some of these proteins are listed in Table 1.

### 8.1. Tumor Microenvironment (TME) and Oxidative Stress

Tumor microenvironment (TME) is a dynamic network of cellular and acellular components that surround tumor cells (parenchyma) to form their niche. Cellular components of TME include stromal cells, such as fibroblasts, immune cells and adipocytes. Acellular components of TME include the extracellular matrix (ECM), growth factors, enzymes, cytokines and chemokines [228]. Tumor cells and TME dynamically and reciprocally interact with each other to achieve their unique characteristics and to drive malignant progression [228,229,230]. TME could be exacerbated by a multitude of intrinsic and extrinsic factors such as oxidative stress [229,231,232]. ROS levels are elevated in tumor cells owing to various causes, including upregulation of ROS-producing enzymes or downregulation of antioxidant enzymes (see above), increased basal metabolic activity, mitochondrial dysfunction, and peroxisome activity [232]. The increased ROS levels not only cause damage in macromolecules such as DNA strand break, DNA-protein crosslinks, and protein degradation [233], but also stimulate further production of ROS in TME. These mechanisms underlie the tumor supportive nature of TME and facilitate tumor progression [229,231]. In response to the increased level of ROS, cellular components of TME, especially, cancer-associated fibroblasts (CAFs) and tumor-associated macrophages (TAMs), not only secrete various proteins that modulate TME, but also further produce ROS [88,123,229,231,232]. Many of these proteins are targeted for S- nitrosylation, aiming to regulate the redox status of TME [231,234]. Among these, the major TME regulators are matrix metalloproteinases (MMPs) and tissue transglutaminase (TG2).

### 8.2. Matrix Metalloproteinases (MMPs) 

MMPs are a family of endopeptidases, containing a Zn^2+^ in their catalytic site [14,126,127]. They play key roles in ECM remodeling, wound healing, morphogenesis and host defense mechanism through their ability to cleave ECM and non-ECM substrates [126,127,235]. In particular, overactivation of MMP-2, -3, -9 and -13 is implicated in carcinogenesis [127]. Upon ROS production by CAFs and TAMs, MMPs are proteolytically activated [229,232]. MMPs then cleave ECM proteins and degrade cell surface molecules, such as E-cadherin and tissue transglutaminase. This causes ECM breakdown and dissociation of epithelial cells, promoting the mobility of cells in TME. MMPs also activate chemokines, cytokines (e.g., CXCL5, CXCL8) and growth factors (e.g., TGFβ) and degrade immune receptors (e.g., the IL2 receptor), which altogether helps establish a pro-tumorigenic TME [235,236]. In particular, MMP9 promotes tumor growth, migration, as well as tumor-associated inflammation. MMP-9 cleaves and releases gelatin and type IV collagen, which compromises the ECM integrity and allows cancer stem cells to invade the ECM by binding to released ECM proteins [237]. In contrast, MMP-9 also inhibits angiogenesis by releasing anti-angiogenic fragments, angiostatin and endostatin, attesting to the complexity of MMP9 functions [235,238].

Cysteine residues in the conserved pro domain of MMP-9 interact with the Zn^2+^ ion, keeping the enzyme in an inactive form. These cysteine residues are targeted for S-nitrosylation. However, the consequences appear to be dependent on NO concentration. S-nitrosylation of these cysteines by low concentrations of NO exposes the Zn^2+^ ion and activates MMP-9 to promote tumorigenesis, a phenomenon termed the cysteine switch [126,127,227,235,239]. In contrast, S-nitrosylation of these cysteines by higher concentrations of NO (for example, during inflammation) is proposed to contribute to the release of the Zn^2+^ ion from the active site and inactivation of MMP-9 [18,126,236,239]. Although the detailed underlying mechanism has yet to be clarified, such S-nitrosylation-mediated inhibition of MMP-9 bears a therapeutic potential for cancer.

### 8.3. Tissue Transglutaminase (TG2) 

Transglutaminases (TGs) are a family of multi–functional enzymes that catalyze transamidation of proteins in a Ca^2+^ dependent manner [120,240]. Among the nine members of TG family, tissue transglutaminase (TG2) is a versatile enzyme expressed in many different types of cells in TME, including smooth muscle cells, endothelial cells fibroblasts and TAMs. TG2 influences tumor growth, metastasis, apoptosis, and chemo-resistance [120,121,241]. TG2’s activities are based on its subcellular localization. Membrane-bound TG2 hydrolyzes GTP and mediates signal transduction for cell cycle progression. In contrast, secreted or cytosolic TG2 crosslinks proteins either by forming stable N-ε-(γ-glutamyl) lysine isopeptide bonds or by incorporating polyamines [120,121,241,242,243]. TG2 crosslinks different ECM proteins, such as collagen, fibronectin and osteonectin, to stabilize the ECM architecture [240,242,243]. TG2 also induces TGFβ activation, leading to the deposition of ECM proteins [121,242]. Furthermore, TG2 elicits anti-apoptotic functions through Caspase 3 inhibition, NF-kB activation and crosslinking (inhibition) of retinoblastoma protein (pRb) during nuclear translocation [240,242,243].

TG2, harboring 18 cysteine residues, undergoes poly S-nitrosylation in a Ca^2+^ dependent manner (241). S-nitrosylation of Cys277 is proposed to inhibit its transamidation activity [121,243]. Since TG2 is implicated in tumorigenesis, this S-nitrosylation-mediated inhibition of TG2 could be utilized in anti-cancer therapy.

## 9. Conclusions

Cumulative evidence suggests that S-nitrosylation plays a large part in NO-mediated biological activities [11]. On one hand, basal S-nitrosylation takes place at the physiological, steady-state level of NO and contributes to the maintenance of homeostasis [10,15,31,197,244,245,246,247]. On the other hand, S-nitrosylation could be induced as a sensor of an oxidizing condition, such as hyperoxia or increased ROS levels. The nitrosyl group could travel to different subcellular locales through a cascade of transnitrosylation reactions [58]. S-nitrosylation is an emerging paradigm of redox signaling that could simultaneously control a large number of proteins involved in divergent pathways. S-nitrosylation helps stop further ROS production, protect cellular proteins against oxidative damage, and propagate redox signaling. The homeostatic level of S-nitrosylation is controlled by the balance between S-nitrosylation and denitrosylation, which are spatially and temporally regulated. However, S-nitrosylation levels are dysregulated in many different types of diseases, such as neurodegenerative diseases and cancer, contributing to the disease pathogenesis. Whether restoration of the homeostatic level of S-nitrosylation could serve as an effective therapeutic intervention for certain diseases warrants active investigation.

## Figures and Tables

**Figure 1 antioxidants-08-00404-f001:**
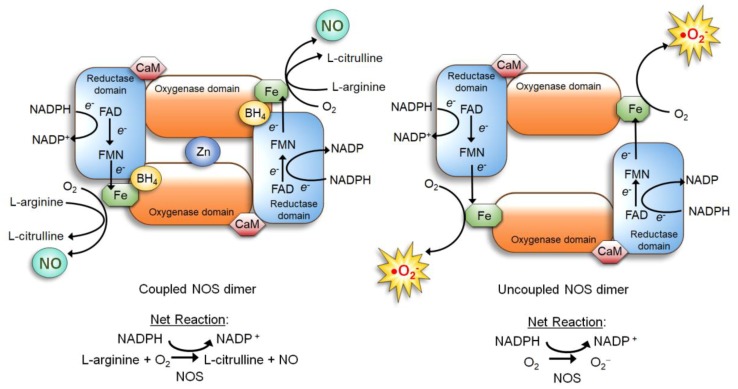
Nitric Oxide Synthase (NOS) dimer in the coupled (normal) vs. uncoupled states [40]. (**Left**) In the normal coupled state, two NOS monomers are tethered by BH_4_, which also increases the binding affinity of the substrate L-arginine. Furthermore, the NOS dimer is stabilized by a zinc ion coordinated in the oxygenase domains of two monomers. Dimerization allows the coupling of two reactions: 1) electron flow: nicotinamide adenine dinucleotide phosphate (NADPH)→ flavin adenine dinucleotide (FAD)→ flavin mononucleotide (FMN)→ Fe of the heme→ molecular oxygen; and 2) oxidation of L-arginine. This coupling yields NO and L-citrulline. (**Right**) When the BH_4_ level is scarce, NOS remains as a monomer and fails to bind L-arginine. The electron flow to molecular oxygen (Reaction 1) is uncoupled to L-arginine oxidation (Reaction 2), yielding superoxide (O_2_•^−^).

**Figure 2 antioxidants-08-00404-f002:**
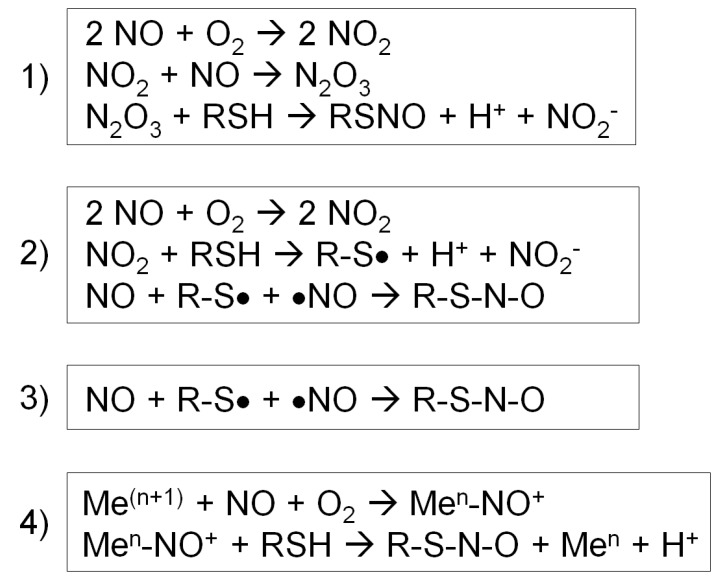
Four different types of S-nitrosylation reactions [56]. (1) Nitric oxide (NO) reacts with O_2_ to form a series of nitrogen oxides. N_2_O_3_ reacts with a protein thiol to produce nitrite and a nitrosothiol. (2) NO reacts with O_2_ to form NO_2_, which reacts with a thiol to produce a thiol radical and nitrite. Then, NO reacts with a thiol radical to form a nitrosothiol. (3) In the presence of a thiol radical, NO directly reacts with it to form a nitrosothiol. (4) NO becomes oxidized by a transition metal, forming nitrosonium (NO^+^). Nitrosonium then reacts with a thiol close to the catalytic center to form a nitrosothiol.

**Figure 3 antioxidants-08-00404-f003:**
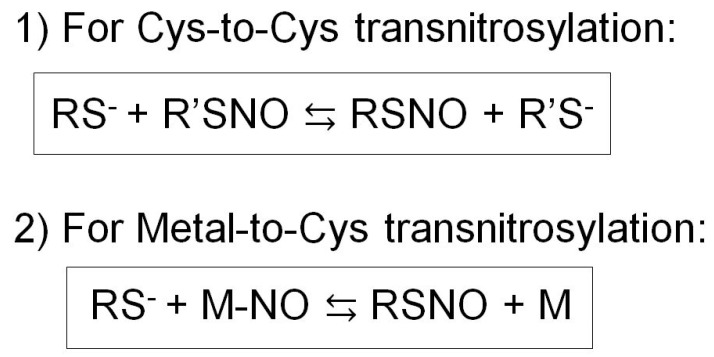
Two types of transnitrosylation reactions. (1) Cys-to-Cys transnitrosylation. (2) Metal-to-Cys transnitrosylation.

**Figure 4 antioxidants-08-00404-f004:**
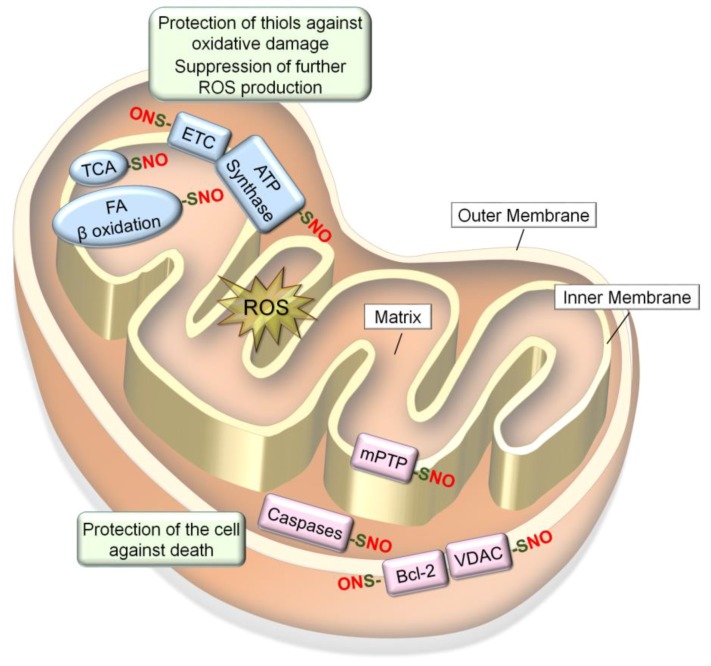
S-nitrosylation of mitochondrial proteins confer resistance to oxidative stress. Mitochondrial proteins become S-nitrosylated in oxidizing environments. S-nitrosylation of proteins involved in respiration and energy production (Electron Transport Chain (ETC), adenosine triphosphate (ATP) synthesis, the citric acid (TCA) cycle and fatty acid β oxidation) leads to prevention of ROS production and protection of protein thiols in these enzymes against oxidative damage. In contrast, S-nitrosylation of proteins involved in cell death regulation (mPTP, caspases-3/-9, VDAC and Bcl-2) leads to protection of the cell against death signals.

**Figure 5 antioxidants-08-00404-f005:**
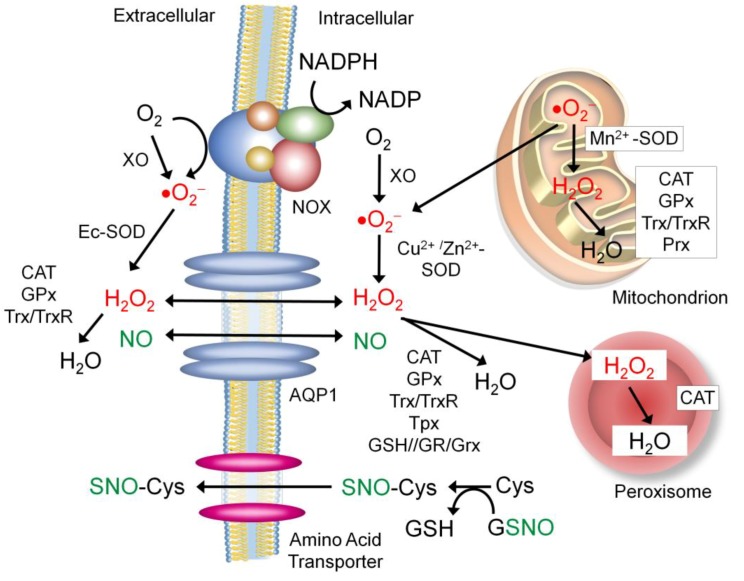
Trafficking of the extracellular and intracellular ROS and NO. Abbreviations: CAT: catalase, Cu^2+^/Zn^2+^-SOD: copper/zinc superoxide dismutase, Ec-SOD: extracellular superoxide dismutase, SNO: S-nitrosocysteine, GPx: glutathione peroxidase, GSH: glutathione, GSNO: S-nitrothiolglutathione, GR: glutathione reductase, Grx: glutaredoxin, Mn^2+^-SOD: manganese superoxide dismutase, NOX: NADPH oxidase, Prx: peroxiredoxin, Trx: thioredoxin, TrxR: thioredoxin reductase, and XO: xanthine oxidase.

**Table 1 antioxidants-08-00404-t001:** Effect of S-nitrosylation of proteins associated with Tumor Microenvironment.

Protein Name	Effect of S-Nitrosylation	Consequence	Ref.
**Bcl-2**	Inhibition of ubiquitination (activation)	Anti-apoptosis	[18]
**Caspases**	Inhibition of caspase activity	Anti-apoptosis	[19,119]
**Fas Receptor**	Promote Fas ligand mediated apoptosis	Apoptosis	[18]
**GAPDH**	Nuclear translocation	Apoptosis	[105]
**TG2**	Inhibition of activity	Tumor suppression	[120,121]
**MGMT**	Inhibition of DNA repair activity	DNA damage & apoptosis	[18,122,123]
**OGG1**	Inhibition of DNA repair activity	DNA damage & apoptosis	[122]
**HDAC2**	Inhibition of deacetylase activity	Histone acetylation	[124,125]
**MMPs**	Activation (low level) Inhibition (high level)	Tumor progression Tumor suppression	[126,127]
**Caveolin-1**	Inhibition of proteasomal degradation	Tumor progression	[128]
**c-Src**	Activation	Tumor progression	[18]
**Β-catenin**	Proteasomal degradation	Tumor suppression	[129]
**HDM2**	Inhibition of binding to p53	Tumor suppression	[130]
**HIF-1α**	Stabilization of activity	Pro-angiogenesis	[14,18]
**MAT**	Inactivation of enzyme activity	Tumor suppression	[82,131]
**NF-kB**	Inhibition of activity	Tumor suppression	[14]
**PTEN**	Inhibition of activity	Tumor progression	[18,132]
**Ras**	Activation	Tumor progression	[133]

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
