# Peer review of "S-Nitrosylation: An Emerging Paradigm of Redox Signaling"

_antioxidants, 2019, doi:10.3390/antiox8090404_

Round 1

Reviewer 1 Report

This review deals with the process of S-nitrosylation in a very detailed manner and takes as an example many proteins that undergo this modification and the resulting effects. The authors have done a good job in detailing the various aspects of the problem.

Author Response

We truly appreciate the reviewer's comments. 

Reviewer 2 Report

This manuscript is a well organized review article focused on the tissue distribution of NOS and the critical role of NO-induced S-nitrosylation in modulating protein function.

The quality of this submission can be better after a minor revision by adding more information on what factors determine the level of BH4 or BH4 stability and the relationship between BH4 and CaM.

Figure 1 shows a close contact between BH4 and CaM, but why CaM still binds to NOS in the absence of BH4? why CaM is required for the formation of NOS dimers?

Author Response

Thank you very much for your constructive suggestions.  We have addressed your concerns as follows:

1. This manuscript is a well organized review article focused on the tissue distribution of NOS and the critical role of NO-induced S-nitrosylation in modulating protein function.

We appreciate your supportive comments.

2. The quality of this submission can be better after a minor revision by adding more information on what factors determine the level of BH4 or BH4 stability and the relationship between BH4 and CaM.

We appreciate your constructive comments.  We have made revisions to address your comments as follows:

1) We have include a paragraph (Lines 153-160) in the revised manuscript that describe different factors that could determine the level and stability of BH4.

2) We have revised a paragraph (Lines 120-135) as well as Figure 1 to show the spatial and functional separation of CaM and BH4. CaM binds the linker region that connects two domains of each NOS monomer.  In contrast, BH4 helps assemble NOS dimer.

3. Figure 1 shows a close contact between BH4 and CaM, but why CaM still binds to NOS in the absence of BH4? why CaM is required for the formation of NOS dimers?

We appreciate your constructive comments. We have revised Figure 1 to clearly show the spatial separation between BH4 and CaM.  CaM binds to the linker region between two domains of each domain, which is independent of dimerization. We have revised corresponding text accordingly.